# Contribution of the Type III Secretion System (T3SS2) of *Vibrio parahaemolyticus* in Mitochondrial Stress in Human Intestinal Cells

**DOI:** 10.3390/microorganisms12040813

**Published:** 2024-04-17

**Authors:** Nicolás Plaza, Diliana Pérez-Reytor, Gino Corsini, Katherine García, Ítalo M. Urrutia

**Affiliations:** Instituto de Ciencias Biomédicas, Facultad de Ciencias de la Salud, Universidad Autónoma de Chile, Santiago 8320000, Chile; nicolas.plaza@uautonoma.cl (N.P.); d.perez@uautonoma.cl (D.P.-R.); gino.corsini@uautonoma.cl (G.C.); katherine.garcia@uautonoma.cl (K.G.)

**Keywords:** *Vibrio parahaemolyticus*, T3SS2, foodborne illness, mitochondria, cell death

## Abstract

*Vibrio parahaemolyticus* is an important human pathogen that is currently the leading cause of shellfish-borne gastroenteritis in the world. Particularly, the pandemic strain has the capacity to induce cytotoxicity and enterotoxicity through its Type 3 Secretion System (T3SS2) that leads to massive cell death. However, the specific mechanism by which the T3SS2 induces cell death remains unclear and its contribution to mitochondrial stress is not fully understood. In this work, we evaluated the contribution of the T3SS2 of *V. parahaemolyticus* in generating mitochondrial stress during infection in human intestinal HT-29 cells. To evaluate the contribution of the T3SS2 of *V. parahaemolyticus* in mitochondrial stress, infection assays were carried out to evaluate mitochondrial transition pore opening, mitochondrial fragmentation, ATP quantification, and cell viability during infection. Our results showed that the Δ*vscn*1 (T3SS2+) mutant strain contributes to generating the sustained opening of the mitochondrial transition pore. Furthermore, it generates perturbations in the ATP production in infected cells, leading to a significant decrease in cell viability and loss of membrane integrity. Our results suggest that the T3SS2 from *V. parahaemolyticus* plays a role in generating mitochondrial stress that leads to cell death in human intestinal HT-29 cells. It is important to highlight that this study represents the first report indicating the possible role of the *V. parahaemolyticus* T3SS2 and its effector proteins involvement in generating mitochondrial stress, its impact on the mitochondrial pore, and its effect on ATP production in human cells.

## 1. Introduction

*Vibrio parahaemolyticus* is a marine pathogenic bacterium that has become the leading cause of gastroenteritis associated with the consumption of raw seafood worldwide [1]. In 1996, the O3:K6 serotype strain of *V. parahaemolyticus* emerged, now known as the pandemic clone, which was responsible for significant gastroenteritis outbreaks globally [2]. In addition to the presence of thermostable direct hemolysin (TDH) and TDH-related hemolysin (TRH), other virulence factors have been described, such as a Type III Secretion System on chromosome 1 (T3SS1) present in all *V. parahaemolyticus* strains [3]. The T3SS1 has a conserved evolution between species of the *Vibrio* genus, especially in environmental strains, for which its acquisition is attributed to a common ancestor [4]. Although this system is responsible for cellular cytotoxicity in a wide variety of human cell lines, studies in animal models suggest that it has a minor role during infection [5,6]. In contrast, some strains associated with the pandemic clone harbor a phylogenetic distinct T3SS named T3SS2 encoded in the pathogenicity island 7 of *V. parahaemolyticus* (VPaI-7) on chromosome 2 [3,7,8]. Studies in animal models strongly suggest that the T3SS2 is essential for *V. parahaemolyticus* to colonize the intestine causing enteritis and diarrhea in the host [5,6,9]. It is for this reason that the T3SS2 is considered the main virulence factor of the pandemic strain of *V. parahaemolyticus*. To date, eleven *V. parahaemolyticus* T3SS2 effector proteins that can be translocated to the eukaryotic cell cytosol have been identified (VgpA, VopA, VopC, VopG, VopL, VopO, VopT, VopV, VopY, VopZ, and VPA1380) [6,10,11,12,13,14,15,16,17,18], which contribute to take control of different cell signaling pathways. Notably, the mechanism involved in generating cell death is not fully understood. T3SSs are multiprotein nanomachines that enable the direct delivery of effector proteins from the bacterial cytosol to the eukaryotic cell cytosol, allowing different effects on cells dependent on the repertoire of translocated effectors, which possess the ability to hijack signaling pathways and modify the activity of diverse host cell organelles, including mitochondria, which are involved in crucial cellular functions such as immune response and energy production [19,20,21,22].

Mitochondria play a central role in regulating cellular functions and host responses to bacterial infections. They are dynamic organelles with a double membrane housing electron transport chain complexes and ATP synthase, adapting energy production based on cell needs [23]. Some effects described in bacterial infections are mitochondrial stress, loss of mitochondrial membrane potential (Δψm), depletion of ATP synthesis, high concentrations of reactive oxygen species (mtROS), and mitochondrial calcium overload leading to the opening of the mitochondrial permeability transition pore (mPTP), which activates various programmed cell death pathways [24]. Therefore, it is not surprising that pathogenic bacteria use mitochondrially targeted effectors to control host cell death and immunity pathways, hijacking and subverting many different host molecules and organelles, including mitochondria [25,26].

Surprisingly, despite the above, only a few T3SS effectors that target host mitochondria and impact the mechanisms underlying their activity have been described or characterized. Among them, *Escherichia coli* EPEC/EHEC T3SS effectors target the mitochondrial pathway. The T3SS effector EspF localizes to mitochondria and is associated with mitochondrial dysfunction and tight junction disruption [25]. Additionally, the effector Map is a mitochondrial association protein imported via the classical TOM/Hsp70 import system and causes mitochondrial dysfunction [27]. In *Yersinia pestis*, YopH mediates the mitochondrial-induced cell death of T lymphocytes and epithelial cells through unclear mechanisms [28]. On the other hand, *Salmonella enterica* mediates cytochrome c release and macrophage apoptosis through the SipB activation of caspase-2 [26]. In the genus *Vibrio*, the effector protein VopE secreted by the T3SS of *Vibrio cholerae* is required to induce mitochondrial fragmentation and suppress innate immune responses during the infection of cultured mammalian cells [21]. Although *Vibrio* is generally considered an extracellular pathogen, the *V. parahaemolyticus* T3SS2 mediates bacterial invasion into host cells through the VopC effector, a deamidase that induces membrane disruption, allowing the internalization by non-phagocytic cells [18,29] to then successfully establish an intracellular lifestyle that could contribute to its survival and dissemination during infection [11,30]. Once *V. parahaemolyticus* invades the T3SS2-dependent infected cells, the intracellular population orchestrates its escape from cells by deploying VPA0226, a lipase secreted by the type 2 secretion system (T2SS) contributing to the generation of mitochondrial fragmentation and morphological changes in the mitochondrial network associated with mitochondrial stress during infections [31].

Despite the fact that the contribution of the T3SS2 in *V. parahaemolyticus* has been described as the key to the colonizing of the intestine and causing enteritis and diarrhea [5,6,9], the contribution to mitochondrial stress that leads to cell death is not fully understood. To address this, we aimed to assess the contribution of the *V. parahaemolyticus* T3SS2 to mitochondrial stress that leads to cell death during infection in human intestinal cells. In this study, we found that the T3SS2 of *V. parahaemolyticus* induces sustained opening of the mPTP, altering ATP production and inducing cell death during infection. These observations suggest that this type of mitochondrial stress could be triggering cell death through the T3SS2 during infection.

## 2. Materials and Methods

### 2.1. Bacterial Strains and Growth Conditions

All strains of *V. parahaemolyticus* RIMD2210633 and its mutant T3SSs, Δ*vscn*1 (T3SS2+) and Δ*vscn*1 Δ*vscn*2 (lacking both T3SSs or T3SS-), used in this study were donated by Dr. Carlos Blondel [32,33]. Strains were routinely cultured in Luria Bertani (LB) liquid medium or on LB agar plates at 37 °C. The culture medium was supplemented with 0.04% bovine and ovine bile (Sigma-Aldrich, Saint Louis, MO, USA, B8381) for the activation of the T3SS2 [34] in infection assays.

### 2.2. Eukaryotic Cell Culture and Maintenance Conditions

HT-29 cells (ATCC HTB-38) were maintained in Dulbecco’s modified Eagle medium (DMEM) (Gibco) supplemented with 10% Fetal Bovine Serum (FBS) (Gibco) (DMEM-10% FBS) at 37 °C in 5% CO_2_. Cells were grown at 37 °C with 5% CO_2_ and routinely passaged at 70 to 80% confluence.

### 2.3. Infection Assay with V. parahaemolyticus Strains

In order to carry out all our experiments in mPTP, mitochondrial fragmentation, ATP determination and cell survival assays, each bacterial strain to be evaluated was cultured overnight and the next day diluted 1:100 into LB liquid media containing 0.04% bile (to induce T3SS2 expression) and grown for 2 h until attaining an OD_600_ of 0.6. Cells were infected at an MOI of 1 and incubated at 37 °C with 5% CO_2_ at each time point assayed (1, 2, 3, 4 and 5 h post-infection (hpi.)).

### 2.4. Mitochondrial Permeability Transition Pore (mPTP) Assay

For the mPTP assay, HT-29 cells were seeded at 2.0 × 10^5^ cells/well into 24-well plates overnight in complete media. Cells were infected at an MOI of 1 and incubated at 37 °C with 5% CO_2_. After infection, the cells were washed with HBSS 1X, the medium was replaced with fresh complete Fluorobrite DMEM medium (Thermo Fisher Scientific, Waltham, MA, USA, A1896701), and an Image-iT LIVE Mitochondrial Transition Pore Assay Kit (Thermo Fisher Scientific, I35103) was used to acquire fluorescence images with Lionheart-FX microscopy (Agilent BioTek, Santa Clara, CA, USA) at different time points of infection. This kit utilizes 1 µM of Calcein-AM (green), which accumulates in the mitochondria of live cells, and 1 mM of Cobalt (Co^2+^) to quench the signal of Calcein when the mPTP is open for a prolonged period. As an experimental control, we use 0.5 µM of ionomycin, an ionophore that induces the opening of the mitochondrial pore. The kit was employed according to the manufacturer’s specifications, and cells were fixed with 4% formaldehyde (Thermo Fisher Scientific, 28908) and the samples were mounted using Prolong Diamond Antifade Mountant (Thermo Fisher Scientific, P36961).

### 2.5. Mitochondrial Fragmentation Assay

For the mitochondrial fragmentation assay, HT-29 cells were seeded at 2.0 × 10^5^ cells/well into 24-well plates overnight in complete media. Cells were infected at an MOI of 1 and incubated at 37 °C with 5% CO_2_. After the infection, the cells were washed with HBSS 1X, the medium was replaced with fresh complete Fluorobrite DMEM medium (Thermo Fisher Scientific, A1896701), a 0.2 µM Mito Tracker Red CMXRos (Thermo Fisher Scientific, M7512) was used for 15 min, the cells were fixed with 4% formaldehyde (Thermo Fisher Scientific, 28908), and the samples were mounted using Prolong Diamond Antifade Mountant (Thermo Fisher, P36961) to visualize HT-29 mitochondria morphology with Lionheart-FX microscopy (Agilent BioTek) at different time points of infection.

### 2.6. ATP Determination Assay

For the ATP determination assay, HT-29 cells were seeded at 2.0 × 10^4^ cells/well into 96-well plates overnight in complete media. Cells were infected at an MOI of 1 and incubated at 37 °C with 5% CO_2_. After infection, the cells were washed with HBSS 1× and cells were lysed using a lysis buffer (HBSS 1× with 1% Triton-X100) to determinate the ATP concentration using The ATP Determination kit (Thermo Fisher Scientific, A22066), a bioluminescent assay designed for the quantitative determination of ATP. It utilizes recombinant firefly luciferase and its substrate D-luciferin. The assay relies on the luciferase dependence on ATP to produce light, with an emission maximum at 560 nm. The resulting supernatant is then used to measure ATP concentrations at different time points of infection. ATP levels are expressed as concentrations in picomoles (pmol) per cell.

### 2.7. T3SS2-Dependent Cell Death Assay

To evaluate the cell viability during infection, HT-29 cells were seeded at 2.0 × 10^5^ cells/well into 24-well plates overnight in complete media. Cells were infected at an MOI of 1 and incubated at 37 °C with 5% CO_2_. After each time point of infection (1, 2, 3, 4 and 5 hpi.), the medium was replaced with fresh complete DMEM supplemented with 100 μg/mL gentamicin to kill extracellular bacteria. Following overnight incubation, surviving cells were quantified using trypan blue exclusion (0.4% trypan blue) and counted on a hemocytometer (Neubauer cell chamber).

To evaluate T3SS2-mediated cytotoxicity, we used the CellTox Green kit (Promega, Madison, WI, USA, G8741) through infection kinetics. HT-29 cells were seeded at 1.0 × 10^4^ cells/well into 96-well plates overnight in complete media. The cells were then washed with HBSS 1× and the Green Dye was added to the suspension at a final concentration of 1×. Cells were infected at an MOI of 1 and incubated at 37 °C with 5% CO_2_. After infection the fluorescence of each condition evaluated was measured at 520nm with the Tecan Infinite M200 Pro Nanoquant at different time points of infection and the results were expressed as a Relative Fluorescence Unit (RFU).

## 3. Results

### 3.1. V. parahaemolyticus Induces mPTP Opening in a T3SS2-Dependent Manner in Intestinal Cells

To determinate the contribution of the T3SS2 of *V. parahaemolyticus* in mitochondrial stress and cell death, we used a double mutant strain of RIMD2210633 lacking both T3SSs (Δ*vscn*1 Δ*vscn*2, T3SS-) and a strain containing functional T3SS2 (Δ*vscn*1, T3SS2+), as previously described [33], in an infection assay with HT-29 cells to evaluate the prolonged opening of the mPTP during infection (Figure 1).

First, we performed a validation of the mPTP assay with imagen analysis by fluorescence with Calcein-AM, a cell permeable fluorophore that diffuses and gets trapped in all subcellular compartments such as mitochondria, in combination with treatment with a cobalt solution (Co^2+^), which quenches calcein fluorescence [35]. To evaluate the mPTP in the uninfected condition, the HT-29 cells were treated with Calcein-AM, Co^2+^ solution, and ionomycin, a known ionophore to induce mPTP opening (stress condition), and contrasted with cells without ionomycin (closed pore or healthy condition) (Figure 1A). Quantification of the intensity was carried out to compare these two states of the mitochondria, producing a decrease in intensity of Calcein-AM in the cells treated with ionomycin, due to the entry of cobalt into the mitochondria or quenching of mitochondrial calcein fluorescence, to demonstrating the opening of the pore (Figure 1B).

Once the mPTP opening assay was validated, we assessed the contribution of the T3SS2 of *V. parahaemolyticus* to the process of opening or closing the pore. In the fluorescence analysis, the infection of HT-29 cells with the Δ*vscn*1 strain (T3SS2+) led to a noticeable quenching of green fluorescence, starting from 1 hpi and progressing over time (Figure 1C). However, when we infected the cells with a mutant strain unable to utilize any T3SSs (Δ*vscn*1 Δ*vscn*2), they showed no fluorescence of Calcein-AM differences compared to uninfected cells (Figure 1C). Quantitative analysis of calcein fluorescence intensity from the images of 150 cells was performed. We observed that the quenching of green fluorescence increases over time, these results being significant in comparison to uninfected cells (Figure 1D). These findings suggest that *V. parahaemolyticus* can induce T3SS2-dependent mitochondrial stress by maintaining prolonged mPTP opening during infection.

### 3.2. V. parahaemolyticus Has the Ability to Induce Mitochondrial Fragmentation, Disrupt ATP Production, and Trigger T3SS2-Dependent Cell Death during Infection

The maintenance of mitochondrial morphology and function relies on a delicate equilibrium between mitochondrial fusion and fission. Mitochondrial fragmentation can occur under cellular stress conditions, manifesting through an escalation in fission activity, an inhibition of fusion, or a combination of both mechanisms [36]. To investigate the role of *V. parahaemolyticus* T3SS2-induced mitochondrial fragmentation, we performed an infection assay and microscopically assessed whether mitochondria network and morphology were changed. Changes in their structure, such as fission or separation and swelling of the cells, are signs of mitochondrial fragmentation (Figure 2). The mitochondrial morphology of the cells infected with a functional T3SS2 strain (Δ*vscn*1, T3SS2+) was determined by the phenotype of the mitochondria labeled with the mitotracker. These were characterized by more rounding than uninfected controls or double mutant strains (Δ*vscn*1 Δ*vscn*2, T3SS-), and in addition, a greater rounding and swelling morphology can be observed between fluorescent dots, which suggested an increased level of mitochondrial fragmentation (marked with the white arrows) (Figure 2A). To quantify these observations, we manually analyzed 50 cells per point from three independent assays and determined the number of cells exhibiting single rounding and swollen mitochondrial morphology. This analysis allowed us to determine the percentage of cells showing mitochondrial fragmentation. In Figure 2B, it is observed that cells treated with ionomycin as a control condition, a known inducer of mitochondrial fragmentation, present a significant increase in the percentage of cells with fragmented mitochondria. For its part, the infection with the functional T3SS2 strain (Δ*vscn*1, T3SS2+) showed a significant percentage of fragmentation from 2 hpi and with progressive accumulation in comparison to the uninfected or double mutant strain (Δ*vscn*1 Δ*vscn*2, T3SS-) (Figure 2B).

Considering that the main source of energy in the cell comes from the mitochondria, we aimed to assess whether the mitochondrial stress induced by the T3SS2 could disrupt the normal ATP production during the infection process. Our observations revealed that in HT-29 cells infected with the strain possessing a functional T3SS2, there was an increase in ATP production by bioluminescence assays at 3 and 4 hpi, significantly higher than initial ATP levels, followed by a decline at 5 hpi (Figure 3A). This is suggestive of classical cellular behavior in the process of cell death, as it has been reported that an increase in ATP production with a subsequent decrease in ATP production is the classic behavior during cell death [37]. On the other hand, the double mutant strain (Δ*vscn*1 Δ*vscn*2, T3SS-) did not exhibit alterations in ATP production during infection, displaying a comparable effect to uninfected cells (Figure 3A).

As observed, *V. parahaemolyticus* can induce early mitochondrial stress in a T3SS2-dependent manner during infection (Figure 2A,B). This is particularly relevant considering previous reports partially linking *V. parahaemolyticus*-induced cell death to the T3SS2 functionality [33,38]. In our case, we performed an infection kinetics test to evaluate cell viability by exclusion of trypan blue, and we also evaluated the integrity of the membrane using the CellTox Green kit to detect the DNA released into the supernatant, in order to verify viability of the cells and what was observed for mitochondrial stress and ATP generation.

We observed that the contribution of the T3SS2 in the generation of cell death was consistent with recently reported data. As expected, HT-29 cells infected with a *V. parahaemolyticus* double mutant strain (Δ*vscn*1 Δ*vscn*2; T3SS-) did not show differences in cellular viability during the infection process, similar to the uninfected cells (Figure 3B). In turn, fluorescence due to DNA release was not increased in the infection process (Figure 3C). In contrast, a functional *V. parahaemolyticus* T3SS2 (Δ*vscn*1, T3SS2+) decreases cellular viability (Figure 3B) and induces the release of eukaryotic DNA (reduction of membrane integrity) from infected cells, a phenotype comparable to cells treated with H_2_O_2_ as a classical inducer of cell death (Figure 3C). These viability results are consistent with what was observed by other authors, where the contribution of the T3SS2 was shown as a significant decrease in cell viability in human intestinal cells [15,32].

All these results suggest that the T3SS2 of *V. parahaemolyticus* induces changes in the morphology and networks of mitochondria, leading to fragmentation and swelling in the early stages of infection, and these alterations have consequences on ATP production at 3 and 4 hpi, promoting cell death during the infection.

## 4. Discussion

In the present work, we have investigated the contribution of the T3SS2 in mitochondrial stress that leads to cell death during *V. parahaemolyticus* infection, being the first report that supports the possibility that the T3SS2 effector proteins are a determinant for generating mitochondrial dysfunction and important for the pathogenesis of *V. parahaemolyticus*. Interestingly, mitochondria are intimately involved in the regulation of intracellular Ca^2+^ fluxes and contain a refined molecular machinery that precipitates regulated cell death via mPTP-dependent regulation [39].

To evaluate the contribution of the *V. parahaemolyticus* T3SS2 to the sustained opening of the mPTP (as a mitochondrial stress signal), we employed fluorescence microscopy using HT-29 loaded with Calcein-AM and Co^2+^ after infection. Calcein-AM is a cell permeable fluorophore that diffuses and gets trapped in all subcellular compartments, including mitochondria [35]. Treatment with Co^2+^ quenches Calcein fluorescence in all subcellular compartments except the mitochondrial matrix, which is enclosed by a Co^2+^-impermeable inner mitochondrial membrane when the mPTP is closed. Thus, the ability of Co^2+^ to quench mitochondrial Calcein fluorescence only when the mPTP is open allows determination of the open vs. closed status of the mPTP in the cell [35]. In our experiments, we observed that the effect of the *V. parahaemolyticus* T3SS2 over mitochondrial health demonstrated the ability to induce mitochondrial stress through mPTP opening (Figure 1). Other pathogens such *Mycobacterium tuberculosis* also generate mPTP-dependent dissipation of Δψ_m_ leading to necrosis in macrophages [40].

The central role of mitochondria in the host’s response to bacterial infections has become a fundamental area of research because they play critical roles in regulating energy production, proinflammatory response, defense against pathogenic infections, and cell death [23,41,42]. To evaluate the contribution of the *V. parahaemolyticus* T3SS2 in perturbing ATP production by mitochondrial stress, we employed fluorescence microscopy to assess mitochondrial fragmentation (mitochondrial fission) and ATP quantification by bioluminescence using human intestinal HT-29 cells at different times post infection. Our results showed that the T3SS2 of *V. parahaemolyticus* contributes to generating mitochondrial fragmentation (Figure 2) and alterations in ATP production, particularly generating an increase in ATP production and then decreasing it over time (Figure 3A). Our results have been consistent with the data reported for *Listeria monocytogenes* that causes dramatic alterations of mitochondrial dynamics via listeriolysin-O (LLO), generating mitochondrial fragmentation induced by *Listeria* infection [43,44]. Furthermore, has been reported that *Legionella pneumophila* abrogates oxidative phosphorylation (OXPHOS) and then, through unknown mechanisms, enhances cellular glycolysis, thereby promoting a metabolic shift known as the Warburg effect [45]. This alteration favors bacterial replication, possibly by reducing the production of antibacterial mitochondrial reactive oxygen species (ROS) [45]. In the genus *Vibrio*, it has been reported that *V. cholerae* also induces mitochondrial bioenergetic dysfunction by ROS generation through its virulence factors GbpA and cholix toxin [46,47]. On the other hand, mitochondrial dynamics (fission and fusion processes) are essential for metabolic activity, where it has been well established that the mitochondrial fusion process generates the greatest efficiency of OXPHOS and increases the ATP production, while on the contrary, mitochondrial fission would decrease the ability to generate ATP [48]. Based on this background, we can observe that our results are similar, since we have shown that the T3SS2 of *V. parahaemolyticus*, in addition to generating prolonged opening of the mPTP (Figure 1), generates mitochondrial fission (Figure 2) and a temporary increase in the production of ATP from the infected cells, before its production declines over time (Figure 3A). In the case of *V. parahaemolyticus*, it has been described that once *V. parahaemolyticus* invades the T3SS2-dependent infected cells, the intracellular population orchestrates its escape from cells by a lipase, VPA0226, that is secreted by the type 2 secretion system (T2SS), contributing to the generation of mitochondrial fragmentation and morphological changes in the mitochondrial network associated with mitochondrial stress during infections [30,31]. In this case, while a mechanism for generating mitochondrial stress by the VPA0226 lipase that depends on the intracellular population of *V. parahaemolyticus* and therefore on the T3SS2 is proposed, a new mechanism for generating mitochondrial stress by the T3SS2 and its effector proteins is proposed in this work. This new mechanism is supported by our results obtained on the contribution of the T3SS2 in generating sustained mPTP opening, mitochondrial fragmentation, and alterations in ATP production (Figure 1, Figure 2 and Figure 3A). Furthermore, these observations are attributable to the T3SS2 and its effector proteins since our results show that using a *V. parahaemolyticus* VopC mutant strain (thus, unable to internalize infected cells) but with its active T3SS2 (T3SS2+), it continues to show sustained mPTP opening and mitochondrial fragmentation (Appendix A).

The control of mitochondrial health during infection that leads to cell death has already been reported in other pathogenic bacteria. *E. coli* EHEC secretes the T3SS effectors Map and EspF, which are targeted to the mitochondria though mitochondrial targeting sequences (MASV) to disrupt morphology, perturb calcium homeostasis, and trigger apoptosis [49]. On the other hand, the T3SS effector protein VopE of *V. cholerae* contains a mitochondrial targeting sequence at its amino terminus that allows precise targeting to mitochondria by hijacking the mitochondrial import machinery [21]. Interestingly, VopE interacts with the GTPase Miro at the mitochondrial outer membrane that was found to perturb mitochondrial perinuclear clustering, which is needed to activate MAVS-mediated NF-kB signaling, an important contributor to the host’s inflammatory response [21].

Particularly, it has been reported that the T3SS2 of *V. parahaemolyticus* contributes to generating cell death during infection [19,20,21,22]. However, to date, the mechanism involved has not been completely understood, such as the process that leads to cell death and the type of cell death generated. In the context of bacterial infections inducing mitochondrial stress, alterations in ATP synthesis and a loss of Δψm leading to mPTP opening triggers the activation of various programmed cell death pathways [24]. Furthermore, mitochondrial stress during the infection of various bacterial pathogens can lead to cell death, including sustained opening of the mitochondrial transition pore and mitochondrial fragmentation causing ATP production alterations [50,51,52]. It has been shown that two different types of cell death occur in Jurkat cells, with the type being determined by the intracellular ATP concentration [52]. The authors observed that the concentration of intracellular ATP acts as a molecular switch controlling the type of cell death. Indeed, they reported that a high concentration of intracellular ATP led to apoptotic cell death, while a low concentration of intracellular ATP led to necrotic cell death. A high level of ATP is generated by cells in response to stress for the activation of proteins, such as caspases, and DNA repair, necessary conditions for apoptosis generation [37]. In contrast, necrosis is characterized by a significant decrease in ATP generation responding to electron transport chain uncoupling [53]. These features of necrotic cell death are similar to those shown by our results (Figure 3), where the T3SS2 of *V. parahaemolyticus* generates first an increase and then a decrease in ATP concentrations (Figure 3A), generating a decrease in cell viability (Figure 3B) and damaging the integrity of the cell membrane from 3 hpi (Figure 3C). This was evaluated by an increase in the signal of a fluorescent probe that can enter cells with a compromised membrane, suggesting that the T3SS2 of *V. parahaemolyticus* would generate mitochondrial stress that leads to necrotic cell death.

In summary, our work identifies a possible role of the *V. parahaemolyticus* T3SS2 and their effector proteins in generating mitochondrial stress that leads to cell death during infection. Particularly, it generates a sustained opening of the mitochondrial transition pore and a depletion in the generation of cellular ATP, similar to necrotic cell death. Future molecular studies on the possible T3SS2 effector proteins involved are required and will be valuable to define the role of these virulence factors in the pathogenicity of *V. parahaemolyticus*.

## 5. Conclusions

Overall, our results indicate that the T3SS2 of *V. parahaemolyticus* is essential to generate mitochondrial stress that leading cell death during infection. To the best of our knowledge, this report proposes a possible role for the T3SS2 of *V. parahaemolyticus* and its effector proteins as a new mechanism for mitochondrial stress generation that leads to cell death during infection.

## Figures and Tables

**Figure 1 microorganisms-12-00813-f001:**
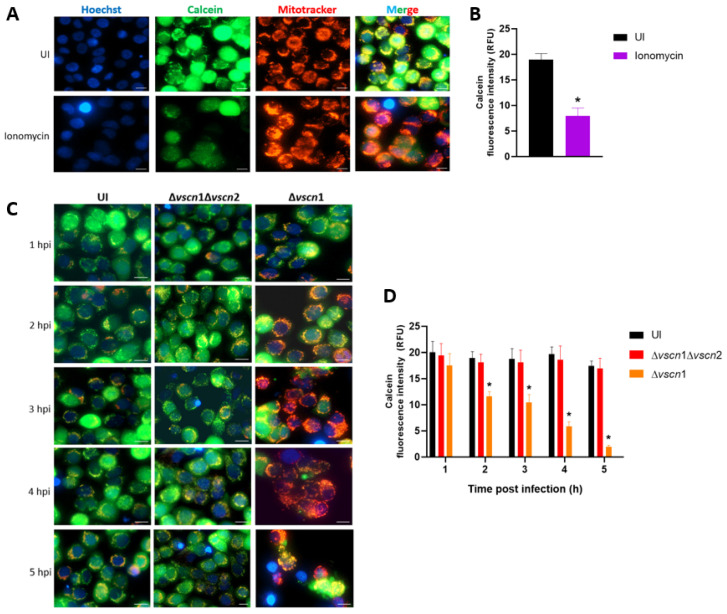
*V. parahaemolyticus* induces T3SS2-dependent mPTP opening in intestinal cells. (**A**,**C**) The HT-29 cells were treated with ionomycin (an mPTP opening inducer), uninfected as a control condition (UI) or infected with *V. parahaemolyticus* strains at different hours post infection. In the figures, the green color indicates the mPTPs are closed (healthy mitochondria), the red color shows the mitochondrial mass (stress condition), and the blue color represents the Hoechst-stained nuclei. (**B**,**D**) indicate the mean fluorescence intensity units of the green channel. Representative images from three independent trials are shown. Statistical significance of differences in calcein fluorescence comparing different conditions with uninfected cells was determined by a one-way ANOVA with Dunnett’s test (*: *p* < 0.05). Scale bar: 10 µm.

**Figure 2 microorganisms-12-00813-f002:**
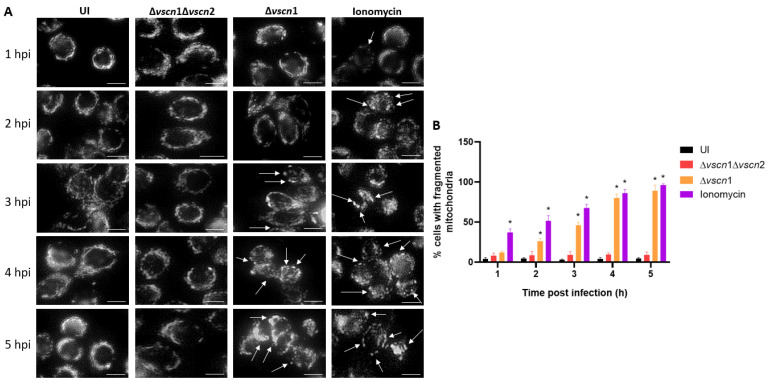
T3SS2-dependent mitochondrial fragmentation during *V. parahaemolyticus* infection. HT-29 cells were infected with different *V. parahaemolyticus* strains or uninfected as a control condition. (**A**) Mitochondria network morphology was analyzed by microscopy. White arrows show cells with mitochondrial fragmentation. (**B**) Percentage of cells with fragmented mitochondria as described for panel A. Representative images from three independent assays are shown. Statistical significance of differences in the percentage of cells with fragmented mitochondria was determined using a one-way ANOVA with Dunnett’s test (*: *p* < 0.05; each condition was compared with uninfected cells at the corresponding times). Scale bar: 10 µm.

**Figure 3 microorganisms-12-00813-f003:**
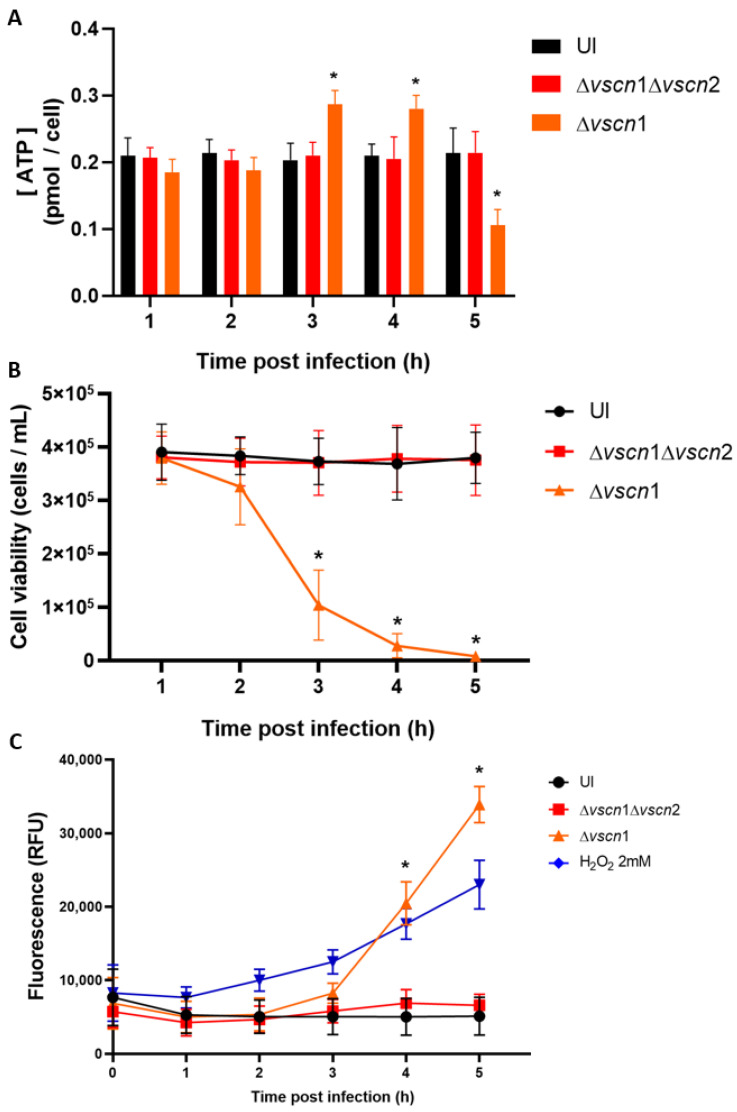
ATP production and T3SS2-dependent cell death during *V. parahaemolyticus* infection. HT-29 cells were infected with different *V. parahaemolyticus* strains or uninfected to evaluate ATP production and cell death. (**A**) The ATP assay was conducted using a bioluminescent assay in cell lysates at the end of each hpi. The values were normalized based on total cell viability. (**B**) Cell viability was evaluated by trypan blue exclusion assay at the indicated times post infection. (**C**) Membrane integrity was evaluated by fluorescence tests at the indicated times post infection. Statistical significance comparing different conditions with uninfected cells at corresponding times was determined by a one-way ANOVA with Dunnett’s test (*: *p* < 0.05).

## Data Availability

Data are contained within the article and in the Appendix A.

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
