# Peer review of "Contribution of the Type III Secretion System (T3SS2) of Vibrio parahaemolyticus in Mitochondrial Stress in Human Intestinal Cells"

_microorganisms, 2024, doi:10.3390/microorganisms12040813_

Round 1

Reviewer 1 Report

Comments and Suggestions for Authors

In this manuscript, Nicolás et al. reported the contribution of V. parahaemolyticus T3SS2 to cause mitochondrial stress during infection. The authors found that the T3SS2 of V. parahaemolyticus induce sustained opening of the mitochondrial transition pore, altering ATP production and inducing cell death during infection. This is a sound investigation, where the data the authors provided can support the conclusions. Comments are given for the authors' consideration.

Comments

1.      All experimental results in this manuscript are presented according to the time post-infection (1, 2, 3, 4 and 5 hpi). If these results correspond to each other, I think it is best to explain the infection conditions (MOI, etc.) in each experimental method.

2.      Only the uninfected control group was added to the results, but I think it would be beneficial to add additional WT/T3SS1+ controls to show more comprehensive data.

3.      The authors proposed that their results were similar to necrotic cell death, not apoptosis. This can be verified by some additional experiments (such as double staining with Hoechst and PI).

Minor comments

1.      Fig.1A: It is written in the legend that DAPI is used to stain nuclei, but Hoechst is marked in figure.

2.      Line 157: (P :< 0.05).

Author Response

Dear reviewer, first of all, thank you very much for taking the time to review this manuscript and for your comments that have undoubtedly improved this manuscript.

Regarding your comments, you can find the answers to your comments below:

1.- Indeed, each of these points corresponds to each experiment conducted under identical experimental conditions. This aspect will be elucidated in the experimental methods section by better segregating the infection process and providing detailed clarification for each experiment.

2.- This is a very interesting point about using a T3SS1+ strain. To date, the contribution of T3SS1 in the virulence of V. parahaemolyticus has been reported in several publications using different cell lines, including human intestinal cells. While it has been reported that T3SS1 can generate rapid cell death at 1.5 hpi, showing a completely different temporality in terms of the phenotypes displayed. On the other hand, only a cytotoxic effect has been attributed by T3SS1 in cell culture, while in the case of T3SS2 both a cytotoxic and enterotoxic effect has been reported in neonatal rabbit models, for these reasons T3SS2 has been considered as the main virulence factor of V. parahaemolyticus, and the main target of study in our laboratory.

Undoubtedly, the contribution of T3SS1 itself is an important point, and we have incorporated in the introduction information that contextualizes the contribution of T3SS1 and the differences mentioned with T3SS2.

3.- This is a very important point and we greatly appreciate the comment. To date, we have not been able to fully propose a mechanism of how V. parahaemolyticus generates cell death, nor have we been able to fully understand the role of T3SS2 effector proteins in the specific cell death mechanism.

In principle, this work seeks to demonstrate the capacity of T3SS2 to generate mitochondrial stress leading to cell death and thus in a future work we will be able to deepen in which T3SS2 effector proteins are involved in cell death elucidating the specific mechanism. To date, only 11 effectors have been described within T3SS2 which contribute to take control of different cell signaling pathways. Notably, the mechanism involves generating cell death is not fully understood and there is a large number of uncharacterized genes encodes within Pathogenicity Island VPAI-7.

We believe that the title could be too ambitious, so we propose to eliminate the cell death part of the title and keep it only with the contribution of T3SS2 in mitochondrial stress. The results in Figure 3 anyway will allow us to keep the discussion on how mitochondrial stress could play a key role in the generation of cell death, being our projections as a research group.

On the other hand, all the minor comments have already been incorporated into the manuscript.

Reviewer 2 Report

Comments and Suggestions for Authors

In this study, the authors systematically investigate the role of the role of the Type III Secretion System 2 (T3SS2) of Vibrio parahaemolyticus in inducing mitochondrial stress and subsequent cell death in human intestinal HT-29 cells. The topic is highly relevant to understanding the pathogenic mechanisms of Vibrio parahaemolyticus, a significant causative agent of gastroenteritis linked to shellfish consumption. The study employs range of assays, including mitochondrial permeability transition pore assays, mitochondrial fragmentation analysis, ATP determination assays, and cell death assays, to investigate the impact of T3SS2 on mitochondrial function and cell viability.

The topic of this article is interesting. To date, the impact of Vibrio parahaemolyticus on human health has not been thoroughly investigated and explored. This article on the impact of Vibrio parahaemolyticus could constitute an advance in terms of the results obtained.

However, there are still some points requiring additional arguments:

1.     The study provides valuable insights into the role of T3SS2 in inducing mitochondrial stress and cell death in human intestinal cells. However, it would be beneficial if the study further explored the underlying molecular mechanisms by identified specific T3SS2 effectors involved in this process.

2.     The use of only one cell line (HT-29) limits the generalizability of the results. Additional experiments using different intestinal cell lines or primary cells could provide a more comprehensive understanding of T3SS2's effects

3.     It would we greatly encourage to evaluate the mitochondrial membrane potential (Δψm). As mitochondrial potential is an indicator for mitochondrial health and Δψm can also trigger different cell death pathways, including apoptosis and necrosis.

Minor comments

1.     There is no scale bar in microscopic images. 

2.     Line 102- mitotracker- Mito Tracker

3.     Specify the Mito Tracker.

4.     Overnight Incubation with 100 μg/mL Gentamycin can be toxic effect against bacteria, Please comment on this.

5.     Line 15. "However the specific mechanism by which T3SS2 induces cell death remain unclear..." (Abstract) should be "remains unclear."

6.     "To evaluate this infection assays were carried out..." (Line 18) - Better phrasing would be needed.

7.     other virulence factors have been described such a Type III Secretion System" should be "such as a Type III Secretion System."

8.     The sentence structure and punctuation in the citations within the text are inconsistent, e.g., references are sometimes mentioned with brackets and sometimes not, which could be standardized for consistency.

9.     Line 140. The term "mitochondria’s" in the section about mPTP assay validation should be "mitochondrias" as the possessive form is not needed.

10.  The phrase "mitochondria involved in crucial cellular functions" could benefit from being rephrased for clarity to "mitochondria, which are involved in crucial cellular functions," adding clarity and readability.

Comments on the Quality of English Language

The document contains numerous grammatical errors that significantly impact its readability and professional quality. It is highly recommended that the manuscript undergo thorough proofreading by an English language expert to ensure clarity.

Author Response

Dear reviewer, first of all, thank you very much for taking the time to review this manuscript and for your comments that have undoubtedly improved this manuscript.

Regarding your comments, you can find the answers to your comments below:

1.- This is a very important point and we greatly appreciate the comment. To date, we have not been able to fully propose a mechanism of how V. parahaemolyticus generates cell death, nor have we been able to fully understand the role of T3SS2 effector proteins in the specific cell death mechanism.

In principle, this work seeks to demonstrate the capacity of T3SS2 to generate mitochondrial stress leading to cell death and thus in a future work we will be able to deepen in which T3SS2 effector proteins are involved in cell death elucidating the specific mechanism. Currently only 11 effectors have been described within T3SS2 which contribute to take control of different cell signaling pathways. Notably, the mechanism involves generating cell death is not fully understood and there is a large number of uncharacterized genes encodes within Pathogenicity Island VPAI-7.

We believe that the title could be too ambitious, so we propose to eliminate the cell death part of the title and keep it only with the contribution of T3SS2 in mitochondrial stress. The results in Figure 3 anyway will allow us to keep the discussion on how mitochondrial stress could play a key role in the generation of cell death, being our projections as a research group.  

On the other hand, we have incorporated additional information both in the introduction and in the discussion on the contribution of other effector proteins that will enrich this work. While V. parahaemolyticus has long been considered an extracellular pathogen, a couple of papers have been reported that V. parahaemolyticus manages its own internalization by using VopC effector protein and survive intracellularly by VopL effector protein within infected cells in a T3SS2-dependent manner. On the other hand, once the bacteria are inside the cells T3SS2-dependent manner, secretes a lipase via a distinct secretion system, the T2SS, to finally escape from the infected cells. That same work reported that T2SS lipase can generate mitochondrial fragmentation. That work and this manuscript suggesting that V. parahaemolyticus could have more than one mechanism to generate mitochondrial stress.

For this, we also propose to add a supplementary image where we use a vopC mutant strain (therefore a strain that cannot internalize) showing that it also generates the mitochondrial transition pore opening and mitochondrial fragmentation, demonstrating that there would be a contribution of T3SS2 and their effector proteins, a mechanism that will be part of a new work as a group.   

2.- Although the mechanism used by V. parahaemolyticus and T3SS2 effector proteins to generate cell death has not been fully understood to date, several studies have reported the contribution of T3SS2 in the virulence of this bacterium in several cell lines independently, including human intestinal cells such as HT-29, Caco-2, showing in all these studies a consistency in the results shown. However, thanks to your comments we have incorporated information into the introduction to support these observations and improve the comprehensive understanding of the effects of T3SS2.

3.- Effectively an important point is the loss of mitochondrial membrane potential (Δψm) and its association with the generation of apoptosis. As we mentioned above, the specific mechanism of cell death and the T3SS2 effectors involved in this process will be part of our future study for ours. On the other hand, we can mention that the experiments performed in this work were carry out with Mito Tracker, who depends on the mitochondrial membrane potential. Therefore, this suggests that the mitochondrial membrane potential would not be altered during infection.

The objective of the cell death experiments (Figures 3B and 3C) is to enrich the discussion considering as plausible the fact that mitochondrial stress would generate T3SS2-dependent cell death, and as a whole our results suggest a more necrotic than apoptotic pathway, by the same fact that mitochondrial membrane potential is not lost, and cell lysis is generated. For this reason, and as mentioned above, it is likely that the original title of this work was too ambitious, which is why it has been proposed to modify the title by evaluating only mitochondrial stress.

Regarding the use of gentamicin in the infection assays, the objective is to kill the bacteria used in the infection assays, after the infection times, in order to subsequently evaluate the viability of the infected cells. This is a modification of the strategy known as gentamicin protection assay.

On the other hand, all the minor comments have already been incorporated into the manuscript.

Round 2

Reviewer 2 Report

Comments and Suggestions for Authors

After a thorough review and careful consideration of the revisions submitted, I encourage editor to accept this manuscript for publication. The authors have effectively addressed the key concerns raised during the review process and have made significant improvements to the clarity and depth of the manuscript. The study provides valuable insights into the role of the Type III Secretion System (T3SS2) of Vibrio parahaemolyticus in inducing mitochondrial stress and subsequent cell death, which contributes to our understanding of its pathogenic mechanisms.

Comments on the Quality of English Language

While the scientific content of your manuscript is strong and the study is of significant interest, there are still several grammatical errors that need to be addressed to ensure the manuscript meets the publication standards